# The psychosocial and economic impacts on female caregivers and families caring for children with a disability in Belu District, Indonesia

**Gregorius Abanit Asa**[1]*, **Nelsensius Klau Fauk**[2,3], **Paul Russell Ward**[2], **Lillian Mwanri**[2]

**1** Sanggar Belajar Alternatif (SALT), Atambua, Nusa Tenggara Timur, Indonesia, **2** College of Medicine and Public Health, Flinders University, Adelaide, South Australia, Australia, **3** Institute of Resource Governance and Social Change, Kupang, Nusa Tenggara Timur, Indonesia

* gorisasa@yahoo.com

**Data Availability Statement:** All relevant data are within the manuscript and its Supporting Information files.

## Abstract

The current study aimed to understand psychosocial and economic impacts of female caregivers and families caring for children with a disability in Belu district, Indonesia. A qualitative inquiry employing one-on-one in-depth interviews was used to collect data from participants (n = 22). Data analysis was guided by a framework analysis for qualitative research. Social implications framework and the economic consequence of disease and injury framework were used to guide the conceptualisation, analysis and discussion of the findings. Findings indicated that female caregivers of children with a disability experienced significant psychosocial challenges. These included feeling frustrated, sad, angry, worried, inferior and insecure due to rejection of their children by other kids with no disability. Poor physical conditions of and negative labelling given to their children and the fear of what the future held for their children with a disability added yet another layer of psychosocial challenges experienced by these women. Separation or divorce and reduced social interaction and engagement in the community were expressed social impact loaded to these women resulting from poor acceptability of the children by their fathers, increased time spent caring and discriminatory and stigmatising attitudes against their children with a disability. The participants also experienced economic impacts, such as increased health and transport expenses, loss of jobs and productivity, and lack of savings. The findings indicate the need for programs and interventions addressing the needs of mothers or female caregivers and families with disabled children. Further studies with large number of participants covering mothers, fathers and caregivers to understand broader experiences and the need of caring for children with a disability are recommended.

## Introduction

The prevalence of people with a disability worldwide is reported to have increased from 10% in 1970 to 15% in 2018 [1]. The 2011 World Health Survey estimated that about 780 million (15.6%) people aged 15 years and older experience disability [1]. Of these, more than 2% have

**Funding:** The author(s) received no specific funding for this work.

**Competing interests:** The authors have declared that no competing interests exist.

problems in functioning. The same survey showed that more than 90 million (5.1%) children in the world are estimated to live with a disability, and of whom, 0.7% experience 'severe disability' such as blindness and severe depression [1, 2].

In Asia and the Pacific regions, 650 million people are estimated to live with a disability, meaning one in every six persons lives with a disability [3]. The number is predicted to increase by virtue of population ageing, poor working condition and chronic health condition, among other factors [3]. South-East Asia is the region with the highest prevalence of moderate disability among other regions, accounting for 16.4% [3]. This indicates that the region might face high level of growth of people with a disability in the next centuries. Indonesia is one of the countries in South-East Asia region where the prevalence of people with disabilities continues to increase. The 2012 national socioeconomic survey reported that the percentage of people with a disability in the country increased from 1.38% in 2006 to 2.45% in 2011 [4]. In 2017, same survey reported that over 37.1 million people live with a disability, and about 10% (3.2 million) are children [5].

Caring for a child or children with a disability within family may cause various challenges to parents, caregivers and other family members [6–8]. Psychosocial issues that face parents and caregivers of children with a disability include stress, feeling of guilt, low self-esteem, negative emotions and behaviours [9–18], worrying about the future of the children, anxiety and depression [17, 19–21] and mental fatigue [22]. Broader impacts on the family include marital dissatisfaction, isolation [23–25], stigma [26–28] and reduced social interaction with other community members [29–31]. Financial implications of having children with a disability include reduced family income [15, 32], significantly high financial demands [25–27, 33–35] and poor access to material and services, which affect not only the parents but other members of the extended family such as siblings and grandparents [36–38] since they contribute in caring for a child with a disability.

Although a range of impacts of caring for children with a disability on families have been reported in different settings, recent systematic literature reviews have highlighted a lack of research on the economic impact of children with a disability in families in resource-poor settings [39, 40]. As poverty was reported as being both a cause and consequence of a disability in childhood, inconsistent evidence exists of the association between socioeconomic status of family and disability in childhood [39, 40]. This study aims to fill the gap in knowledge and to explore the economic impact of caring for children with a disability among mothers or female caregivers in resource poor settings. In addition, psychosocial impacts were also explored to add new knowledge and to provide a comprehensive understanding of the lived experiences and impacts of childhood disability on health and general wellbeing of mothers or female caregivers in this study and in Indonesia where evidence on the impacts of childhood disability is still limited [41, 42]. In many resource-poor settings, mothers are responsible for caring for children, husbands and other household members [43, 44]. It is thus possible that caring of children with a disability may increase burden of caring on mothers. Understanding the psychosocial and economic impacts of caring for children with a disability on mothers or female caregivers in families is important for policies and programs addressing disability impacts facing families. The current study aimed to understand psychosocial and economic impacts experienced by mothers or female caregivers and families with children with a disability in Belu district, Indonesia.

## Methods

Consolidated criteria for reporting qualitative studies (COREQ) checklist [45] was used to guide the report of the methods section of this study. The COREQ checklist contains 32 required items (S1 Fig) for explicit and comprehensive reporting of qualitative studies especially interviews and focus groups [45].

## Theoretical framework

The economic consequence of disease and injury [46], and social model of disability [47] guided the conceptualisation, analysis and discussion of the study findings. The economic consequence of disease and injury framework explains three possible economic impacts experienced by female caregivers and families caring for children with a disability, including increased health expenditure, labour and productivity losses, and financial capital. This framework suggests that ill-health has negative impacts on family economic condition through increased family health expenditures due to the increased need for healthcare or treatment. Health expenses may be paid out of current income or from cash savings, if available, or via loan or the sale of family assets [46]. The reduction family income, savings and assets may in turn lead to reduced investment in physical, financial and human capital. The framework also suggests that ill-health influences family economic condition through labour and productivity losses due to increased time spent to look after the sick family member(s) [46].

The social model of disability guided the presentation of several social impacts of disability on family [47]. This model suggests that disability is a social construction where society imposes hinderances to the full participation of people with a disability. The hinderances are reflected in different negative or discriminatory attitudes and behaviours against people with a disability within communities where they live and interact [47]. The model also shows that society unjustly privileges certain appearances and levels of functioning as normal over others, resulting in social exclusion, and economic and political marginalization [47]. Social exclusion and barriers can also lead to financial hardship, stress and anxiety among families caring for member(s) with a disability [47]. Thus, the ones who do not conform to the expected appearances, behaviours and levels of functioning are considered to have subnormal status and inferior within society [47, 48].

## Study setting

The study was conducted in Belu district, East Nusa Tenggara, Indonesia. The district has a total population of 204,541 [49] and covers the area of 1,284.97 km2. It consists of 12 sub districts and shares the border with East Timor. The economic growth is dominated by the agricultural sector, followed by forestry and fishery sectors. The population growth is 3% per year [49]. There is a state special school for children with a disability (*Sekolah Luar Biasa Negeri*, also known as SLBN) located in the district, which provides education for children with a disability at elementary school (44 students), junior high school (14 students) and senior high school (13 students) level. There are also two private rehabilitation centres for children with a disability which are called *Pusat Rehabilitasi Hidup Baru* run by sisters of Franciscan and *Bhakti Luhur* run by the sisters of ALMA (Asosiasi Lembaga Misionaris Awan) Congregation. The two rehabilitation centers each takes care of 27 and 31 children with a disability, respectively. Belu is reported as one of the districts in East Nusa Tenggara province with the highest number of children living with a disability, accounting 348 children [50]. To the best of our knowledge, there is also limited research evidence illuminating the impacts of childhood disability on parents or caregivers in Indonesia [41, 42]. Because of small size, feasibility, familiarity and the potential of undertaking the current study successfully, Belu was selected as the study setting.

## Study design and data collection

A qualitative design was employed in this study. Qualitative research has been found to be useful in exploring issues in settings or situations that participants face in their daily living. The use of qualitative design in this study was considered appropriate for the discovery and

explanation of issues and the views of participants [51–53], and for exploring psychological and socioeconomic impacts on mothers or female caregivers and families caring for children a disability.

Data collection occurred from August to September 2019, using face-to-face in-depth interviews with mothers and female caregivers of children with a disability. The inclusion criteria for participants recruitment included a woman aged 18 years old or older, who were caregiver of a child with a disability at a state special school for children with a disability and two rehabilitation centres in the study setting. Participants were recruited using a combination of purposive and snowball sampling techniques. At the first stage, the researchers enlisted the help from the principal of the state special school for children with a disability and the leader of the rehabilitation centers for people with a disability to distribute the study information sheet to mothers or female caregivers of children with a disability. Those who contacted and agreed to participate in this study were recruited and interviewed in a private room at the rehabilitation centres or school, and at participant-researcher mutually agreed time. The interviews were conducted by the first author (male) who is a freelance researcher and has attended a formal training in qualitative methods during his master's study. Interviews were guided by several predetermined main questions and probing questions were developed alongside the progress of each interview to allow the participants to communicate much more freely and to provide more detailed descriptions or information about the topics being investigated [54–56]. The decision about the questions was made through a process of formulation, discussion and revision by the first and second authors, and based on the conceptual frameworks used in this study and the study aim [57]. Interviews focused on several key areas, including psychosocial and economic related impacts of caring for children with a disability on the participants (S2 Fig). Initial participants were also asked to disseminate the study information sheet to their eligible friends and colleagues who might be willing to take part in the study. This process was recursive, and the recruitment stopped once the researcher felt that the collected information was rich enough to explain the topic being studied and data saturation had been reached. A total of 22 participants were interviewed and there was no participant withdrawal or repeated interviews with any participants. The interviews were carried out in Bahasa with the duration ranging between 30 to 45 minutes and recorded using a digital recorder. Notes were also taken during the interviews. Only the researcher and participant were present in the interview room. There was no established relationship between the researcher and participant prior the interview. At the end of the interviews, each participant was offered an opportunity to read and correct the recorded information after the transcription, but none required to do so.

## Data analysis

The recorded data were transcribed and translated into English by the first two authors (GAA and NKF) and checked for accuracy by other authors (PRA and LM). To maintain the quality and validity of the data, cross check and comparison of the data were performed during the transcription and translation process. Data were imported into software package NVivo 12 for analysis which was guided by Richie and Spencer's framework analysis [58]. The framework comprised five steps as follows: (i) *familiarisation* with the data through listening to the audio recordings, reading transcripts, breaking down into several chunks of data, giving comments and labels and listing key ideas; (ii) *identifying a thematic framework* by making judgements about the importance and relevance of key issues and concepts from participants. In this stage, the thematic framework was constructed; (iii) *indexing all the data* through highlighting chunks of the text and analyse open codes to look for similar or redundant codes and reduce them into smaller number as codes referring to the same theme were grouped together to

reach a few overarching themes and sub-themes; (iv) *creating a chart* where the indexed data were summarised and arranged in charts that corresponded to the thematic framework; (v) *mapping and interpretation the data* as a whole [53, 58–60]. The interpretation of the findings was undertaken using the above two main theoretical perspectives, including the economic consequence of disease and injury and the social implications framework. A dualistic approach was undertaken whereby these theories helped develop a theoretical lens for deductive analysis to occur, as well as allowing emergence of new theories from the data (inductive analysis) [52, 59, 61].

### Ethical consideration

The study obtained the ethics approval from Health Research Ethics Committee, Duta Wacana Christian University, Yogyakarta, Indonesia (No. 618/C.16/FK/2018). Prior to the interviews, each participant was advised about the aim of the study and that their participation was voluntary and that there would be no consequences if they choose to withdraw their participation during the interview. Participants were also informed that each interview would be recorded using a digital recorder and take approximately 30 to 45 minutes. They were also assured that information they provided would be treated confidentially and anonymously. Each participant was assigned a study identification letter and number (e.g. R1, R2) for de-identification purposes. Before the commencement of the interviews, each participant signed and returned a written informed consent form.

## Results

### Profile of participants

A total of 22 participants aged between 35 and 60 years old were involved in this study. The majority of them were married and biological mothers of the children, two women were caregivers (grandmother and aunty), and four women were single parents. Among the participants, the majority (14) graduated from junior high school, six graduated from senior high school, and two finished elementary school. Most of the participants were unemployed (18 people), two were household assistants (domestic work) and two were shopkeepers. The majority of participants were housewives (18 –given that four were unmarried or widowed) and from poor families as husbands generally earned very low income working as carpenters, street vendors, vegetables and chicken merchants or farmers. The details of the children of these participants are presented in Table 1. Participants reported various impacts of caring for their children with a disability. The findings were grouped into three main themes:(i) psychological, (ii) social, and (iii) economic impacts. Detailed explanation is provided below.

### Social impacts

**Husband-wife separation.** Disharmony within families and husband-wife separation or divorce were identified as the consequences of having a child with a disability. A few women

**Table 1. Characteristics of the children.**

| Type of disability | Number of children | Age range (years) |
|---|---|---|
| Visual impairment | 3 | 8–13 |
| Hearing impairment | 3 | 9–15 |
| Speech impairment | 4 | 6–14 |
| Physical handicap | 5 | 6–14 |
| Cognitive impairment | 3 | 10–15 |
| Mental and physical impairment | 4 | 10–16 |

acknowledged that the presence of a child with a disability was not fully accepted by their husbands or partners and had led to husband-wife separation and divorce:

> "He was 3 years old and we had been doing many treatments for him. But only little positive change occurs. His mother, father and I looked for ways such as going to prayer (pendoa), and traditional treatment (obat kampung). His father was living together with another woman and escaped from our village. Her mother knows that his father does not accept the son with a disability" (R2: 48 years old female caregiver/aunty).

> "Her father and I planned to have a matrimonial sacrament. But everything had changed since our child was born with a disability. He (her father) never talked about legal marriage. I asked him about marriage, but he did not reply. He said he cannot marry me. I said okay and since then I have to work hard for the future of three kids. Her father was not close to her [his child with a disability] and preferred to play with the twins who have no disability. Now he has left us, I have to work harder for the future of all my three kids" (R7: 49 years old mother).

**Reduced social interaction and engagement in community activities.** Caring for children with a disability within families was indicated to limit mothers or caregivers' engagement in social interactions and community activities. Participants comments that they were not able to engage in social or community activities in the community and had to change social behaviour including being quiet, because of the need to supervise or care for, and a loss of trust in others to look after their children with a disability:

> "I used to talk a lot, but not anymore. My behavior changed since she was born with a disability. I do not talk to others or even trust them asking for help as I used to. I use my time for looking after her and thinking a lot" (R11: 39 years old mother).

> "I cannot take part in community activities because everybody in my house is busy with their own things. Sometimes I want to join, but I cannot because I need to watch my son, I do not really trust them [other family members] to look after him" (R16: 42 years old mother).

Discriminatory and stigmatising attitudes and behaviours against children with a disability by other kids and neighbours were also the supporting factors for reduced engagement in social interactions and community activities among the participants. Several participants expressed discomfort around engaging in social or community affairs because of the fear that they may be asked by other community members or neighbours about their children's conditions. They were also unwilling to meet people who negatively labelled their children:

> "I am not feeling comfortable to meet and interact with the others [community members] because they may ask about my child condition" (R22: 45 years old mother).

> "People [community members] are often eager to know what is going on within your family. I am not comfortable to answer questions related to my son conditions. It is better to stay at home and not to interact with them that often" (R21: 52 years old mother).

> "I do not participate in community activities because I do not like to meet those people [community members] who said negative things about my child. I do not like to meet them in community activities because they may look kind and talk nicely to me, but at the back they said bad things about my child" (R20: 39 years old mother).

However, several participants seemed to engage in community activities as a strategy to receive support from other members of the community or governments and to mitigate the potential for negative consequences such as: being neglected by other community members, being ineligible for government aid, and being cynically questioned by others.

*"If we [the woman and her husband] do not participate in community activities, they [other community members] might not come to our house to give a hand when I have an event at home"* (R13: 42 years old mother).

*"They [other community members] know that we have a child disability, and if we are often absent from community activities, we might not receive government aid, such as subsidised rice delivery"* (R9: 39 years old mother).

*"A neighbour will ask cynically why we [the woman and her husband] do not take part in community activities. If we say that we are looking after a family member with disability, they might say 'you need to make time. . ..' To avoid this repetitive question, we just need to go to these activities"* (R5: 42 years old mother).

### Economic impacts

**Increased health and transport expenditures.**   Taking care of children with a disability was also reported to increase economic burden on their family. Increased healthcare expense for children with a disability was one of the challenges:

*"To me, the most challenging issue is the cost of accessing therapy because there is no special therapist here. We decided to undergo therapy in Surabaya but we did not stay there long. My son and I had to come back to Atambua (the study setting) because we ran out of money. One-hour therapy costs IDR 150,000 (±USD12) and a one day's therapy runs for 6 hours, excluding food and accommodation/living costs"* (R11: 39 years old mother).

*"I gave him supplement but there was no change. Then I took him to the doctor and the doctor suggested for us to go to Bali for brain scanning. In my heart I said I cannot afford it. It is so expensive: the flight, living cost and scanning. The normal medical treatment here has already costed us a lot of money. I gave up"* (R17: 48 years mother).

Cost for transportation for daily activities was another economic impact experienced by participants. This seemed to be exacerbated by the lack of permanent jobs, leading mothers to sell their personal items to cover daily commuting cost to school:

*"I have to drop him off and pick him up every day at special school. We live a bit far from his school, so we have to pay IDR 10,000 (±UDS 70 cents) one way, which means we spend IDR 20,000 for transport every day. It is expensive"* (R11: 39 years old mother).

*"His father is a carpenter and I am a housewife. I work part time for a Chinese family: washing dishes and cleaning up the house but not every day. Aci (the Chinese woman in the family) could give me IDR 50,000 or sometimes IDR 10, 000 every time I do the work (The minimum monthly salary for a person with a bachelor's degree in Belu district in 2020 is IDR 1,793,298 (±USD 125.21). Generally, the wage for casual workers is lower than permanent workers and determined by the employers). The motorbike taxi (ojek) cost for my kid is IDR 4,000, meaning IDR 8,000 every day. Sometimes, when we do not have money to pay the ojek I ask him not to go to school, waiting until we have money"* (R1: 46 years old mother).

**Loss of jobs and productivity.** Loss of jobs and productivity seemed to be another economic impact facing mothers or female caregivers and families caring for children with a disability. Female caregivers and also fathers could lose their jobs when they had to focus their attention on a family member with a disability. A few participants stated that they quitted their jobs or reduced working hours due to the dedication they needed to make to care for their children with a disability:

*"Prior to his birth I worked as a non-permanent civil servant. I had to stop working when we knew he had a disability. My uncle wanted me to work again to support the family, but I decided to focus on my son instead"* (R9: 39 years old mother).

*"I used to work 7–9 hours a day, selling vegetables in the market. But since I have a child with disability, I only work 3–4 hours per day"* (R3: 36 years old mother).

Some participants reported to have increased time spent on taking care of their children with a disability, including bathing, feeding at home and commuting to school. This increase in caring activities negatively influenced their involvement in income generating activities:

*"I feed and bath my children at home every day, and my daughter [the one with disability] always wants me to stay and wait for her at school. So, I have to wait for 5 hours every day at school. She cries if she does not see me during break. I cannot do anything else to earn money for us in the family"* (R7: 49 years old mother).

*"It takes 20 minutes to school by ojek and another 20 minutes to get back home. To pick him up from school I have to be at least 40 minutes earlier before he finishes school. So, it takes me about 1.5–2 hours every day to drop off and pick him up from school. I do not even think of working to earn money because the situation is not really supportive"* (R4: 50 years old mother).

**Lack of savings.** Taking care of a child with a disability was indicated to influence families financially and hampered parents' ability to save for their future. The participants acknowledged that it had been difficult for them to save money as they spent all the income they earned for their daily needs and the needs of their children with a disability. Such conditions seemed to be supported and exacerbated by the lack of permanent job opportunities and low payments for the parents or caregivers:

*"It is too far to talk about savings. I work 2 to 3 days per week and sometimes every two weeks. Each day I might get IDR 50,000 or IDR 100,000, depending on employers' kindness. I spend all the income I have for the need of my little one [a child with disability] especially for her transport to special school every day"* (R8: 43 years old mother).

*"Honestly, I have never had any savings for my entire life. I get paid but it is only to support our daily needs for three to four days ahead. Sometimes I have no work during a month, and there is no permanent job that I can find that suits my situation of caring for my child"* (R10: 45 years old mother).

*"He [her child with disability] wants to have a music instrument, but I did not end up buying it because I had spent all the money I previously saved for his needs like medicine and transport expenses and his brother's school fee"* (R18: 48 years old mother).

*"We had a motorbike at home, but then I sold it because his father run away and no one could ride it. I spent all the money from the motorbike sale for the treatment of my son and*

*his transportation cost to school every day. So, I do not have savings" (R1: 46 years old mother).*

## Psychological impacts

**Feeling frustrated, sad and angry.**   Participants faced various psychological challenges including experience of feeling frustrated, sad and angry due to reasons such as: rejection by other children of their children with a disability, and negative labelling given to their children by neighbours and/or by other community members:

*"What makes me frustrated is that neighbouring kids do not want to play with my child. It is so painful. They like to touch my child's toys, but they do not want my child to touch their toys. Other children are very welcome to come to our house, but it is not applied to my kid. They lock the door when my kid is close to their houses. . . .. Sometimes I do not allow my child to play with the them because I feel their attitudes towards my child is unacceptable" (R14: 42 years old mother).*

*"He (my child) often laughs out loud by himself in our veranda and is heard by neighbours. Some parents and their children hearing it say that my kid is crazy or mentally ill. . . I feel angry and want to rebuke them, but then I stand and try to stay calm. . . .. I would not social-ise with neighbours like them" (R15: 41 years old mother).*

*"In October and May every year we have rosary prayer [for catholic religion]. We pray every night moving from one house to another. My son always wants to pray to Mother Marry but he cannot finish it. Every time he prays children around him burst into laughing or giggling. It is sad and stressful. Once I asked the head of the prayer community to skip him so that to another person can pray" (R19: 51 years old mother).*

**Feeling worried of economic condition.**   Participants talked at length about the psychological impact related to their family's finances as a result of their child having a disability. For example, participants acknowledged that they felt stressed out with their economic condition:

*"I feel worried when I do not have money to pay my son's ojek [motorbike taxi] transport from home to school and back to home. My son always asks me why, when I have not taken him to school on the day, and I always have to tell him that today I do not have transport cost" (R1: 46 years old mother).*

*"My husband is a carpenter. If he comes home without any money in his wallet, I feel a bit anxious because I have to think of what household item, I should sell in order to buy my son's medicine or cover transport cost" (R8: 43 years old mother).*

*"I have to work every day because I am a widow, the only breadwinner for my three children. If I am sick, I cannot earn income to fulfill my children needs. This makes me worried especially if I think of the one with disability" (R5: 47 years old mother).*

**Feeling inferior and insecure.**   Feeling inferior was one of recurring themes mentioned by the participants. Such a feeling arose when they compared their life and their children's physical conditions with those of other people or children:

*"I feel like other people in the community are better off as they have good life and normal chil-dren, not like me" (R6: 42 years old mother).*

*"I often feel inferior meeting people or when other people look at my child. I feel like people are judging us because my child has poor physical condition" (R18: 43 years old mother).*

Feeling insecure about life and the future of children with a disability within family was also experienced by mothers and a grandmother caregiver of the children. Some participants acknowledged the possibility of disadvantaged situations such as falling sick, losing jobs and passing away that might occur any time and lead to negative consequences on their families and the future of their children who have a disability. The expression of these feelings can be attested as follows:

*"I have to work hard and do my best at work because I am the only breadwinner and a widow. I feel insecure if I am sick and it will be more insecure if I get fired" (R8: 53 years old mother).*

*"I wash other families' [people] clothes. This is not a stable or secure job. I hope they keep calling me to work for them" (R1: 46 years old mother).*

*"I do not know where this kid's father is. I am her grandmother, and her mother does not work. I think a lot about her future when her mother and I pass away. I do not think anyone will look after her, even family" (R12: 59 years grandmother caregiver).*

## Discussion

The impacts of caring for children with a disability have been reported to affect families in many settings [6–8, 62, 63]. This study aimed to understand the psychosocial and economic impacts on female caregivers and families caring for children with a disability in Belu district, Indonesia. Consistent with other studies [64–68], the findings of the current study suggest that female caregivers of children with a disability experienced psychological burdens such as frustration, sadness and anger due to discriminatory and stigmatising attitudes and behaviours (rejection, mockery and negative labels) of other parents and children against their children with a disability. As the issue of disability has not much been addressed through policies and programs in Belu, it is plausible to argue that stigma and discrimination towards children with a disability seem to be underpinned by the lack of understanding and awareness of community members about disability conditions, negative impacts facing families with children with a disability and supports they need. Economic or the financial hardships experienced within families exacerbated worries, stress and/or psychological challenges experienced by the study participants. These findings support the construct of the economic consequence of disease and injury framework [46], the social model of disability [47], and earlier findings [9–12, 19–22, 24, 69], reporting negative impacts of disability. Based on the findings in this study, it is reasonable to state that increased time spent caring and increased expenses for medical and daily needs of the children with a disability, were the mechanisms through which disability induced negative psychological consequences on the study participants. The current study also suggests that having children with a disability could also lead to participants' feeling inferior especially when they made comparison between their life with those of other people and neighboring children who did not need to care for children with a disability. Such inferior feeling among the participants seemed to be caused by the prevalent disability associated stigma within communities where they lived and interacted. Reduced and low earnings and unreliable job opportunities were also additional issues that enhanced psychological challenges, gravely enhancing caregivers' fear for the future of their children, especially in unforeseen instances of them falling sick.

Consistent with previous studies [23–26], the current findings show that having children with a disability within a family had negative social implications on spousal or marital relations between husband and wife, leading to consequences such as separation and divorce. In contrast to previous studies [70–74], reasons for spousal separation or divorce among the current study participants were not related to challenges of raising and nurturing children with a disability and dealing with their special needs and the needs of family, but non-acceptance behaviour or rejection of the husbands towards the physical conditions of these children. Such rejection may be supported by embarrassment of having a child with a disability due to stigma and discrimination against children with a disability still exist, and inability or lack of understanding of how to raise and deal with a child with special needs. Supporting the previous findings [29–31, 75], the current study suggests that caring for children with a disability limited social interaction and participants' engagement with community. Additionally, some participants felt the necessity to disengage with the community to concentrate on the role of supervision of their children with a disability. This behaviour has also been described in a previous study [75], reporting that being responsible mothers is among the main reasons for low social participation in community activities. Additionally, in line with the construct of social model of disability [47], negative labeling, unreasonable questioning and social rejection towards children with a disability, inferior feeling about the situation around a child with a disability were also factors leading to study participants' disengagement with the community activities and events. Therefore, such social disengagement or withdrawal seemed to also be undertaken by the participants as a strategy to avoid stigma and discrimination against their child with a disability. In conformity with a study elsewhere [76], the role of physical caring of children with a disability (e.g. bathing, feeding, commuting to schools), was expressed as one of the most difficult experiences among study participants.

Supporting previous studies [25–27, 33–35, 77, 78], and the construct of economic consequences of disease and injury framework [46], the current study's participants incurred high healthcare costs, necessitating the cancelation of the medical treatments (e.g. stop buying supplements and brain scans) for some children with a disability. Similarly, relatively high cost of other necessities such as transport to and from school for children with a disability and their carers was a heavy burden on some families leaving no option but to opt out. Such conditions seemed to be exacerbated by the lack of employment opportunities and the participants' low level of education which made them ineligible to compete for limited job opportunities available in the study setting. However, for some women quitting jobs, reducing working hours and limiting engagement in income generating activities were used as strategies to cope with the demanding care and supervision roles, another mechanism through which childhood disability negatively impacted on them and their families economically and socially. Increased time spent caring and increased expenses for medical and daily needs of the children with a disability were the mechanisms through which disability caused negative socioeconomic consequences on the study participants. Thus, it is apparent that the current study's findings present evidence on the association between the economic status of mothers and female caregivers in this resource-limited setting and disability in childhood.

## Limitations and strengths of the study

There are several limitations that need consideration in interpreting the findings. Firstly, like many case studies in qualitative research, the findings of the study may reflect the unique conditions of the participants in Belu district, which would be different to participants caring for children with a disability in different settings. Secondly, we only explored the perspectives of female caregivers at one point in time and did not explore views of other family members.

This may have led to incomplete overview of impacts of disability experienced within families caring for children with a disability. However, to our knowledge this study represents an initial qualitative understanding of impacts of childhood disability experienced by female caregivers and families in the context of Indonesia, and Belu district in particular. Thus, the current study's findings are useful to inform disability-related programs and interventions addressing impacts of disability on parents, caregivers and families in Belu and other similar settings in Indonesia and globally. Future studies on this topic, which involve significant number of mothers and fathers, other family members and caregivers are recommended as they may provide a more complete picture of the impacts of disability in families.

## Conclusions

The study reports several impacts on female caregivers and families having children with a disability in the study setting. Feeling frustrated, sad and negatively labelled around the neighbourhood and feeling inferior, insecure and stressed due economic hardships, were psychological impacts experienced by female caregivers of children with a disability. Spousal separation and divorce, disengagement from social interaction and community activities, were the social implications of childhood disability identified in this study. Families with children with a disability experienced economic hardship due to the loss of jobs and productivity, poor earning and/or savings and increased healthcare and other daily living expenditure. The current findings indicate the need for programs and interventions to address the impacts of disability on parents, caregivers and families. For example, psychological support through counseling for mothers and female caregivers of children with a disability, as it has been considered effective in helping people facing different difficult life situations [79, 80]. Similarly, findings also indicate the need for disability-related awareness raising and education for family and community members in Belu and other settings in the country to increase their knowledge and awareness of disability and acceptance of and support for children with a disability people, which may reduce stigma and discrimination against children with a disability and their family. As economic impact seemed to have significant negative influence on other aspects of the participants and their families, interventions that address the economic aspects of mothers or female caregivers and families of children with a disability in Belu and other similar settings in Indonesia are recommended. Free transport for children with a disability, financial capital aid for small scale business, such as retailer and food stall and support workers for mothers or caregivers of children with a disability would represent examples of interventions helpful in caring for children with a disability and their families. Caring for children with a disability causes a range of psychological, social and economic challenges on parents and families, hence addressing their needs to cope with these challenges through policy and practice is considered necessary.

## Supporting information

**S1 Fig. COREQ checklist.**
(DOCX)

**S2 Fig. Interview guide.**
(DOCX)

## Author Contributions

**Conceptualization:** Gregorius Abanit Asa, Nelsensius Klau Fauk.

**Formal analysis:** Gregorius Abanit Asa, Nelsensius Klau Fauk.

**Investigation:** Gregorius Abanit Asa.

**Methodology:** Gregorius Abanit Asa, Nelsensius Klau Fauk, Paul Russell Ward, Lillian Mwanri.

**Project administration:** Gregorius Abanit Asa, Nelsensius Klau Fauk.

**Writing – original draft:** Gregorius Abanit Asa, Nelsensius Klau Fauk.

**Writing – review & editing:** Gregorius Abanit Asa, Nelsensius Klau Fauk, Paul Russell Ward, Lillian Mwanri.

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
