## [Decision Letter · Decision Letter 0]

15 Sep 2020

PONE-D-20-13748

The psychological and socioeconomic impacts on female caregivers and families caring for children with a disability in Belu District, Indonesia

PLOS ONE

Dear Dr. Asa,

Thank you for submitting your manuscript to PLOS ONE. After careful consideration, we feel that it has merit but does not fully meet PLOS ONE’s publication criteria as it currently stands. Therefore, we invite you to submit a revised version of the manuscript that addresses the points raised during the review process.

We look forward to receiving your revised manuscript.

Kind regards,

Joseph Telfair, DrPH, MSW, MPH

Academic Editor

PLOS ONE

Journal Requirements:

2. Please include additional information regarding the interview guide or script used in the study and ensure that you have provided sufficient details that others could replicate the analyses.

For instance, if you developed a guide as part of this study and it is not under a copyright more restrictive than CC-BY, please include a copy, in both the original language and English, as Supporting Information.

Reviewers' comments:

Reviewer's Responses to Questions

**Comments to the Author**

1. Is the manuscript technically sound, and do the data support the conclusions?

Reviewer #1: Yes

2. Has the statistical analysis been performed appropriately and rigorously? 

Reviewer #1: N/A

3. Have the authors made all data underlying the findings in their manuscript fully available?

Reviewer #1: Yes

4. Is the manuscript presented in an intelligible fashion and written in standard English?

Reviewer #1: Yes

5. Review Comments to the Author

Reviewer #1: The paper presents a qualitative study investigating the psychosocial and financial impact of caring for a child with disabilities in a sample of 22 Indonesia women.

The study contributes to a growing literature about the strains and pressures associated with caregiving, adding the perspective of women from a developing country. The investigation is theoretically justified and the methodology is overall sound. There are however some aspects of the background, methods and conclusions in the study that need improvement in my opinion, as I have listed below:

1. Title and throughout the paper: Given the theoretical frameworks informing this study, with one heavily based on social aspects, I would encourage the authors to consider changing "psychological" to "psychosocial" impact; the study does not seem to be framed within a psychological model, but on a social and economic, thus, even the psychological impact is shaped by that. This is reflected in Theme 1 "feeling frustrated, sad and angry" and Theme 3 "Feeling inferior and insecure" of Psychological impact, which both speak about socially determined psychological states, further supporting the idea that this study teased out psychosocial impact.

2. The Introduction provides information about prevalence of disability in Indonesia, but no figures are presented about the proportion of children who have disabilities. This information would be useful to better contextualise the study.

3. The intro justifies researching the economic impact of caregiving due to a gap in the knowledge, but there is no evident rationale for investigating the psychosocial impact, and the literature discussed in the Introduction appears to be quite extensive on this topic. It would be useful to clarify for the reader why psychosocial impacts were investigated.

4. Study Setting: The authors provide a clear description of Belu, but it would be good to explain briefly why this setting was chosen. Are there differences in prevalence of children with disabilities between this district and the national context?

5. Study design and data collection: It would be useful to have more information about the interviews: were they structured or semi-structured? What was the range of duration of the interview? I would strongly encourage the authors to add the interview schedule or main interview questions in a supplementary file. Also, how were questions decided and by which members of the research team? What equipment was used to record the interviews?

6. Study Design: The main body of the manuscript does not make reference to the COREQ checklist in the supplementary file, which would be important to mention. Importantly, the COREQ checklist should include the page number where each point of the list has been addressed, which has not been done in the submitted document. I also note that some points of the checklist are ticked but have not been addressed, as for instance point 19 on audio/video recording

7. Results: While I agree with the authors about describing the psychological impact separately from the social and economic impact, I would suggest to move this as the final theme, because there are important overlaps between the three themes and the psychological wellbeing of the participants appears to be a result of the social and economic impact. Just a suggestion.

8. The Discussion is overall clear, but I feel that the summary of main findings could be less descriptive and try to go a bit deeper into the mechanisms leading to the psychosocial and financial impact considering the particular social and cultural context. Do we see here an accumulation of social stigma and financial strains that are found in other regions? How much of the findings can be generalised? These aspects are touched upon, but the summary of findings is very long, and the more speculative part is a bit lost.

9. The Conclusions section suggests potential interventions, but it doesn't appear to touch on implications for tackling social/cultural attitudes and stigma. As they say, 'it takes a village', and this study clearly highlights how these women live in communities where they don't feel supported emotionally, psychologically, let alone financially. This is an important point emerging from this study, I feel that it should be given more weight.

6. PLOS authors have the option to publish the peer review history of their article (what does this mean?). If published, this will include your full peer review and any attached files.

Reviewer #1: **Yes: **Marica Cassarino

---

## [Author Response · Author response to Decision Letter 0]

1 Oct 2020

The responses to reviewer's comments are attached as a file.

---

## [Decision Letter · Decision Letter 1]

6 Oct 2020

The psychosocial and economic impacts on female caregivers and families caring for children with a disability in Belu District, Indonesia

PONE-D-20-13748R1

Dear Dr. Asa,

We’re pleased to inform you that your manuscript has been judged scientifically suitable for publication and will be formally accepted for publication once it meets all outstanding technical requirements.

Kind regards,

Joseph Telfair, DrPH, MSW, MPH

Academic Editor

PLOS ONE

Additional Editor Comments (optional):

Reviewers' comments:

Reviewer's Responses to Questions

**Comments to the Author**

1. If the authors have adequately addressed your comments raised in a previous round of review and you feel that this manuscript is now acceptable for publication, you may indicate that here to bypass the “Comments to the Author” section, enter your conflict of interest statement in the “Confidential to Editor” section, and submit your "Accept" recommendation.

Reviewer #1: All comments have been addressed

2. Is the manuscript technically sound, and do the data support the conclusions?

Reviewer #1: Yes

3. Has the statistical analysis been performed appropriately and rigorously? 

Reviewer #1: N/A

4. Have the authors made all data underlying the findings in their manuscript fully available?

Reviewer #1: Yes

5. Is the manuscript presented in an intelligible fashion and written in standard English?

Reviewer #1: Yes

6. Review Comments to the Author

Reviewer #1: All comments were addressed. The manuscript is much more improved.

7. PLOS authors have the option to publish the peer review history of their article (what does this mean?). If published, this will include your full peer review and any attached files.

Reviewer #1: **Yes: **Marica Cassarino

---

## [Editor Report · Acceptance letter]

26 Oct 2020

PONE-D-20-13748R1 

The psychosocial and economic impacts on female caregivers and families caring for children with a disability in Belu District, Indonesia 

Dear Dr. Asa:

I'm pleased to inform you that your manuscript has been deemed suitable for publication in PLOS ONE. Congratulations! Your manuscript is now with our production department. 

Kind regards, 

on behalf of

Dr. Joseph Telfair 

Academic Editor

PLOS ONE